# Liquid Leachate Produced from Vermicompost Effects on Some Agronomic Attributes and Secondary Metabolites of Sweet Basil (*Ocimum basilicum* L.) Exposed to Severe Water Stress Conditions

Hatice Kosem [1], Mehmet Zeki Kocak [2], Mustafa Guven Kaysim [3], Ferdi Celikcan [1] and Muhittin Kulak [2,*]

[1] Department of Organic Farming Management, College of Applied Science, Igdir University, Igdir 76000, Türkiye
[2] Department of Herbal and Animal Production, Vocational School of Technical Sciences, Igdir University, Igdir 76000, Türkiye
[3] Department of Field Crops, Faculty of Agriculture, Igdir University, Igdir 76000, Türkiye
[*] Correspondence: muhyttynx@gmail.com or muhittin.kulak@igdir.edu.tr

**Abstract:** Water stress is one of the most critical threats to the growth and productivity of plants and is one of the most studied topics in agricultural sciences. In order to enhance the tolerance of plants to water stress conditions, synthetic fertilizers have been widely used in the field. However, due to their toxic effects, recent reports have focused on organic options. In this study, the effects of liquid vermicompost applications (25, 50, 75, and 100%) on the agronomic attributes, phenolic compounds, and essential oil compounds of basil plants exposed to drought stress conditions were investigated. Accordingly, water stress critically reduced the factors of plant height, plant fresh weight, root fresh weight, leaf length, and leaf diameter. On the other hand, vermicompost applications significantly affected all of the parameters considered, except the leaf length of well-watered basil plants. However, a two-way ANOVA analysis revealed that the interactions of water stress and vermicompost were significant on root length and root fresh weight. Regarding the essential oil compounds, the contents of humulene, anethol, eucalyptol, estragole, bisabolene, germacrene, and caryophyllene were quantified. Estragole was determined as a major component by 85–90%. The results revealed that the highest estragole content was determined in the 25% vermicompost + water stress, water stress, and control groups. Of the major phenolic compounds, caffeic acid decreased as a result of water stress conditions but increased with vermicompost treatments. The rosmarinic acid content increased during water stress conditions, attaining the highest content at 25% via the vermicompost and water stress interaction. In general, the 25% and 50% vermicompost applications increased the content of phenolic compounds in plants under either well-watered or stress conditions.

**Keywords:** waste management; organic amendment; phenolics; terpenoids; abiotic stress

## 1. Introduction

Plants are exposed to both biotic and abiotic environmental factors throughout their lives; however, relevant environmental factors cause stress to the plant if they exceed the plant's ability to cope with the level of stress. Although the effects of the relevant stress factors depend on the plant species or the severity, type, and duration of the stress, these factors can often delay the growth and development of plants, reduce their productivity, and may ultimately cause their death [1,2].

Of the vital abiotic stress factors that exist, drought stress is one of the major problems for crops. For that reason, drought stress is one of the most widely investigated abiotic stress factors in agricultural sciences. Its deleterious effects have been reported for numerous plant species in general [3,4] and for *Ocimum* sp. in particular [5–8]. The adverse effects manifest as stunted growth and performance, which, in turn, result in a critical loss of crop productivity. As reported in a numerous reviews and research reports [9–12], the slowing

down of plant growth is crucially linked to damage to the photosynthesis system, which subsequently affects assimilate production and its allocation to plant tissues. Corresponding to the perturbations occurring in the plant metabolism process, critical shifts from primer to secondary metabolism processes occur due to a carbon surplus. The relevant shifts are deemed to be the non-enzymatic adaptive strategies of the plants [13–15]. Secondary metabolites are some of the crucial non-enzymatic compounds exerting protective roles against either biotic or abiotic stress elicitors [16,17]. Considering the secondary metabolites (phenolics and terpenoids), *Ocimum basilicum* is a reputed species characterized by a high content of rosmarinic, caffeic, and chicoric acids, as well as methyl chavicol (estragol), eugenol, linalool, methyl cinnamate, and camphor [18,19].

In order to ensure global food security and to make the plants more compatible with water constraints, synthetic or semi-synthetic fertilizers have been employed in the research, and, subsequently, a higher yield of crops has been produced. However, the high levels of chemical inputs have caused the contamination of the soil, water, and air. Due to the residues produced by the relevant agricultural chemicals, adverse effects on human and animal health have also been reported in the literature [20]. These negative effects have led researchers to use organic fertilizers that are compatible with the natural environment and do not pose a toxic threat to other living organisms. One of the organic fertilizers commonly used is vermicompost. Vermicompost is not only an important compost and bio-control factor, but also an effective means of solid waste management. These organic amendments are also crucial for the sustainability of agricultural activities and subsequently food security due to their contributions to the physical, chemical, and biological properties of soil [21].

Vermicompost is an organic fertilizer obtained from food processed in the digestive system of certain waste-eating worm species (*Eisenia fetida, Eisenia andrei, Dendrobaena veneta, Lumbricus rubellus, Perionyx excavatus*) [22]. Vermicompost increases the water holding capacity of soil, increases the plant's resistance by competing with the beneficial bacteria in its structure, is non-toxic, regulates the soil's pH level, and positively affects certain parameters, such as plant fresh and dry weights and yield [23,24]. Considering its effects on basil plants, several reports revealed the affirmative effects of vermicompost on the vegetative growth of basil [25–31]. Enhanced plant performances have been attributed to the regulation of photosynthesis, antioxidant enzyme activity, and secondary metabolites [30,32–35]. Although organic fertilizers, such as vermicompost, are known to have a positive effect on plant growth, the action mechanisms of vermicompost concerned with the physiology and biochemistry of plants are still not fully elucidated in the literature. Similar to the case of secondary metabolism, the research mostly focuses on the essential oil yield and composition of plants under well-watered conditions. Considering the phenolic acids and flavonoids and their alterations against stress conditions, Celikcan et al. [30] reported that the vermicompost-enhanced crop productivity of the plants under well-watered conditions and the relevant amendments might not be effective in coping with water stress conditions. However, the critical changes occurring in phenolics or terpenoid compounds were noted alongside the treatments, but the changes in the secondary metabolites were not manifested or translated into the enhanced tolerance of basil against water stress conditions.

According to our research and knowledge, the effects of a liquid vermicompost fertilizer application to both the agronomic properties and secondary metabolites of basil plants have not yet been studied in the literature. For that reason, the current study aims to investigate the potential uses of vermicompost effluent (leakage) for basil plants grown under water stress conditions. Moreover, organic fertilizers are compatible with the soil structure and plant and provide significant contributions due to their high nutrient and organic matter contents, in general. Corresponding to the uses of organic fertilizers, the nutrient status and organic matter content of the soil lost over time might be maintained/buffered as a result. In the present study, the experimental soils were enriched with vermicompost prior to being subjected to water stress conditions. The hypothesis of the study addresses the enrichment of the soil. Due to the compounds with molecular structure analogues

similar to the hormones, enzymes, elements, and bacterial flora available in vermicompost, we hypothesized that the enrichment of the soil with vermicompost would result in significant changes in the agronomic attributes of the basil plant and in secondary metabolite composition. Due to the contribution of vermicompost to the root system of the plants, we further hypothesized that the vermicompost-mediated development in the root systems of the basil plants would provide a greater tolerance to water stress conditions.

## 2. Materials and Methods

### 2.1. Experimental Site, Plant Materials, Submitting Water Stress, and Harvest Time

The experiment was conducted at the research greenhouses of the Agricultural Research and Application Centre, Igdir University, Türkiye. The study was performed as a factorial experiment using a completely randomized design with three replicates. Sweet basil (*Ocimum basilicum* L.) seeds were purchased from Simagro Agro & Seed Company (Konya, Türkiye). Of the medicinal and aromatic plant taxa, basil plants are one of the preferred species due to their chemical composition characterized by a high-essential-oil and phenolic content. For that reason, we used basil plants for the current study. In this regard, the seeds were initially surface-sterilized using 1% ($v/v$) hypochlorite for 2–3 min, and then the seeds were rinsed with distilled water to remove the residue of the disinfectant. After re-drying the seeds to their original moisture content using a tissue paper at room temperature, the seeds were sown in 2 L plastic pots containing peat and grown in greenhouses for a 14/10 h photoperiod, 26–30 °C/ day and 16–20 °C/ night; relative humidity: 60%. From germination to the final harvest, the irrigation of the control plants was based on the field capacity of the experimental soil. Regarding the estimation of soil water content/pot water capacity, the experimental soils were firstly fully saturated and then the pots were weighed. Subsequently, the soil samples were dried at 105 °C until a constant weight was obtained. The differences between the weight of fully saturated and dried soil samples were quantified, which were considered as the water weight required for the pot water capacity. Regarding the irrigation levels, the pot weights were estimated every second day and then transpiration-mediated water losses were buffered with re-watering to obtain the soil water capacity [30]. Once the basil seedlings grew 6–8 true leaves, the seedlings underwent severe water stress by water-holding for eleven days. The seedlings were susceptible to the water-holding stage after 11 days; for this reason, the experiments were terminated and the relevant samplings were performed following an 11-day drought period. The stress period was based on the wilting point of the plants. Concerned with the vermicompost treatments and their interactions with the stress, prior to submitting them to water stress conditions, the experimental soils were firstly enriched with 25%, 50%, 75%, and 100% concentrations of vermicompost once a week for four weeks. At the end of this period, the plants were exposed to drought stress for 11 days. All the measurements were performed with three replicates and each replicate corresponded to ten plants. The experimental design of the study is presented in Table 1.

**Table 1.** Experimental design of the study.

| Acronym | Vermicompost Treatments | Irrigation Level |
|---|---|---|
| Control | Leachate amended | Well-watered plants |
| Water Stress (WS) * | Non-leachate amended | Severe water stressed plants |
| 25% LVC ** | Leachate amended (25% LVC/75% distilled water, $v/v$) | Well-watered plants |
| 25% LVC + WS | Leachate amended (25% LVC/75% distilled water, $v/v$) *** | Severe water stressed plants |
| 50% LVC | Leachate amended (50% LVC/50% distilled water, $v/v$) | Well-watered plants |
| 50% LVC + WS | Leachate amended (50% LVC/50% distilled water, $v/v$) | Severe water stressed plants |
| 75% LVC | Leachate amended (75% LVC/25% distilled water, $v/v$) | Well-watered plants |
| 75% LVC + WS | Leachate amended (75% LVC/25% distilled water, $v/v$) | Severe water stressed plants |
| 100% LVC | Leachate amended (100% LVC/0% distilled water, $v/v$) | Well-watered plants |
| 100% LVC + WS | Leachate amended (100% LVC/0% distilled water, $v/v$) | Severe water stressed plants |

* WS: water stress; ** LVC: liquid leachate obtained from vermicompost, *** $v/v$: Volume/volume.

## 2.2. Vermicompost Preparation and Physicochemical Properties of Liquid Leachate of Vermicompost

The vermicompost was produced, as we previously reported [30], in dark conditions with (20 ± 2 °C) and 75% humidity. The sources of the vermicompost were cow manure and *Eisenia fetida*. The analysis of the physicochemical composition of relevant liquid leachate was conducted at the Soil, Fertilizer and Water Resources Central Research Institute (Ministry of Agriculture and Forestry, Türkiye). The analysis revealed that the composition was organic matter content (0.44%); pH (6.98); EC (18.12 dS/m); total nitrogen (N) (0.14%); total potassium (K) (0.30%); total copper (trace level; Tr); total phosphorus (P) (0.05%); total calcium (Ca) (0.01%); total magnesium (Mg) (0.01%); total iron (Fe) (Tr); total manganese (Mn) (Tr); and total zinc (Zn) (Tr).

## 2.3. Agronomic Traits

Agronomic traits such as plant height, plant fresh weight, root length, root fresh weight, leaf fresh weight, leaf length, and leaf width, were assayed with ten plants for each replicate with a total of thirty basil plants corresponding to the three replicates.

## 2.4. Solid-Phase Micro-Extraction (SPME) of Essential Oils and GC-MS Conditions

For the extraction of essential oils, the method optimized by [30] was assayed. Briefly, 0.5 g of dried and powdered basil leaves were mixed with 10 mL of double distillate water, and then the relevant mixture was stirred at 45 °C for 30 min. The stirring was followed by the trapping of essential oil using an SPME holder (Supelco 57330-U) needle for a 7-minute period. Then, the trapped volatiles on the SPME holder needle were injected into the GC-MS and left for 4 min on the relevant septum. The analysis for the identification of the essential oil components lasted for 33 min. Each analysis was performed with three replicates. Considering the identification and relevant analysis of the essential oil components, Thermo GC-MS Trace Ultra (USA) was used. Regarding the GC-MS conditions, a DB-5MS column (30 m × 0.25 mm × 0.25 μm) was used and the flow rate of the carrier gas of helium was set as 1.0 mL/min. The oven temperature was kept at 40 °C for 1 min and then increased from 40 to 120 °C at a rate of 5 °C/min and maintained for 2 min. The temperature was then increased to 240 °C with a rate of 10 °C/min and maintained for 3 min. The injection part temperature was set to 240 °C. The mass spectrometer was operated in EI mode at 70 eV. The split ratio was set as 20:1. Mass range: 45–450 m/z; scan speed (amu/s): 1000. The components were identified in comparison to NIST08, Willey7n.1, and HPCH1607 libraries reference compounds.

## 2.5. Extraction and Quantification of Phenolics Using LC–MS/MS

The harvested basil leaves were, firstly, dried and powdered. Then, shaker-aided and sequential extraction were performed at 120 rpm at room temperature for 24 h. In that context, 3 g of basil leaves were extracted using 50 mL of methanol. The extraction was repeated three times with the same plant materials to collect all residues following the extraction. The filtrates of each extraction were then vacuo-dried using a rotary evaporator (Heidolph 94200, Bioblock Scientific, Germany). The extracts were then preserved at +4 °C until chromatographic analysis was performed. For the quantification of the phenolic acid and flavonoids, ultrahigh performance liquid chromatography (Shimadzu Nexera, Kyoto, Japan) coupled with a tandem mass spectrometer (LCMS8040 model) was used. Considering the conditions of LS-MS/MS, similar modified and optimized conditions of [36,37] were applied. This was performed as the reversed-phase UHPLC was equipped with a SIL-30AC model autosampler, a CTO-10ASvp model column oven, LC-30CE model binary pumps, and a DGU20A3R model degasser. Different analytical columns, *viz.*, RP-C18 Inertsil ODS-4 (100 mm × 2.1 mm, 2 μm) and 120 EC-C18 models (150 mm × 2.1 mm, 2.7 μm), were used and the column temperature was set to 40 °C. Methanol and acetonitrile were used as the mobile phases, while ammonium formate, ammonium acetate, acetic acid, and formic acid were used as the mobile-phase additives. The gradient elutions were 20% B (35–45 min), 100% B (25–35 min), and 20–100% B (0–25 min). The flow rate was set to

0.5 mL/min and the injection volume was 5 μL. An ionization source (ESI) was used to perform spectrometric detection. ESI was operated in positive-ionization mode for vanillin, daidzin, piceid, coumarin, and hesperidin, while ESI was operated in negative for other standards. MS conditions: drying gas (N2) flow: 15 L/min; nebulizing gas (N2) flow: 3 L/min; interface temperature: 350 °C; heat block temperature: 400 °C; and DL temperature: 250 °C [36,37].

### 2.6. Experimental Design and Statistical Analysis

The experimental design corresponded to a factorial model in a completely randomized block, with treatments being irrigated/non-irrigated with liquid leachate, and drought-stressed/non-drought-stressed plants. For each measurement, three replications were used, and each replicate corresponded to ten plants. The experimental data were analyzed via two-way ANOVA. The relevant variances were related to major treatments (liquid leachate and water stress conditions) and their interactions. The means were separated using Duncan's multiple range test at a 5% probability level ($p < 0.05$) (SPSS 22). Additionally, heat map clustering was conducted in order to visualize and associate the parameters (ClustVis online). Principal component and correlation analyses were performed using JAMOVI and GraphPad Prism, and a network plot analysis was performed using PAST Software.

## 3. Results

### 3.1. Agronomic Attributes

Water stress significantly affected the agronomic attributes of the sweet basil, as estimated from the shorter plant height and leaf length, taller root length, lighter plant FW and leaf FW, heavier root FW, and smaller leaf width ($p = 0.000$) (Table 2), as was the case commonly observed for sweet basil under a restricted water supply [6,8,38,39]. However, independent from water stress conditions, applications of liquid leachate significantly affected the relevant attributes of the basil, except the root FW and leaf length, as observed from the values corresponding to the taller plant height and root length, heavier plant FW and leaf FW, as well as wider leaf width. Additionally, the impacts of the leachate were concentration-dependent, suggesting an increase from 25 to 50% for the plant height, plant FW, leaf FW, and leaf width, and decrease from 75 to 100%. Furthermore, the root length increased by the increasing concentration of leachate. Considering the interactions of water stress and vermicompost, only the root length and root FW were observed to be significant ($p = 0.000$). Under water stress conditions, root FW substantially decreased with the leachate amendments, whilst root length increased with the treatments, in general (Table 2).

### 3.2. Heat Map Clustering, Correlation, Principal Component, and Network Plot Analyses of the Agronomic Attributes Corresponding to the Treatments

In addition to the two-way ANOVA analysis we performed, the relevant data of the agronomic attributes were subjected to an array of statistical analyses in order to reduce the dimension, and correlate, visualize, and clarify the experimental results corresponding to the treatments. Such analyses are quite common in research with a high number of variables. Firstly, we constructed a heat map. According to the clustering presented in the heat map, water stress and vermicompost treatments were clearly sorted into two distinct clusters. The first cluster included "*control* and *all vermicompost treatments*", corresponding to the *well-watered groups*. On the other hand, the second cluster included "*stress* and *its interaction with vermicompost treatments*", corresponding to the *stress-submitted groups*. The results suggest that irrigation status is a critical predictor with respect to the agronomic attributes. Of the estimated attributes, the under-ground components of the plant, *viz.*, root length and root FW, were clearly separated from the above-ground components of the plants corresponding to the treatments (Figure 1). Additionally, the correlation analysis clearly revealed the negative coefficients between under- and above-ground parts, but a significant correlation was only noted between leaf length and root length ($r = -0.84$) (Figure 2).

**Table 2.** Effects of liquid leachate obtained from vermicompost (25, 50, 75, and 100%) on some agronomic attributes of sweet basil (*O. basilicum* L.) under drought stress conditions.

| Treatments | Plant Height (cm) | Plant FW (g) | Root Length (cm) | Root FW (g) | Leaf FW (g) | Leaf Length (cm) | Leaf Width (cm) |
|---|---|---|---|---|---|---|---|
| Control | 13.480 ± 1.000 cde | 3.900 ± 0.229 d | 14.460 ± 0.841 c | 0.353 ± 0.045 fg | 1.096 ± 0.110 c | 4.020 ± 0.453 b | 1.610 ± 0.079 bcd |
| WS * | 8.333 ± 0.666 f | 2.683 ± 0.480 e | 24.847 ± 1.264 a | 1.024 ± 0.132 bc | 0.520 ± 0.076 d | 2.820 ± 0.072 c | 1.320 ± 0.092 d |
| 25% LVC ** | 16.700 ± 1.410 b | 4.767 ± 0.737 abc | 14.933 ± 0.306 c | 0.747 ± 0.095 cde | 1.550 ± 0.132 b | 4.203 ± 0.300 ab | 1.870 ± 0.066 ab |
| 25% LVC+ WS | 12.517 ± 0.797 de | 4.203 ± 0.211 cd | 22.277 ± 0.751 b | 0.653 ± 0.115 de | 1.253 ± 0.105 c | 2.808 ± 0.357 c | 1.338 ± 0.078 d |
| 50% LVC | 18.933 ± 1.504 a | 5.227 ± 0.261 a | 21.350 ± 1.103 b | 0.910 ± 0.168 bcd | 2.033 ± 0.260 a | 4.230 ± 0.305 ab | 2.080 ± 0.203 a |
| 50% LVC+ WS | 15.333 ± 0.950 bc | 4.457 ± 0.172 bcd | 20.550 ± 0.853 b | 0.790 ± 0.236 b–e | 1.543 ± 0.081 b | 2.967 ± 0.153 c | 1.787 ± 0.220 ab |
| 75% LVC | 15.200 ± 0.900 bc | 5.060 ± 0.333 ab | 14.767 ± 0.751 c | 0.597 ± 0.015 ef | 1.990 ± 0.105 a | 4.660 ± 0.295 a | 1.867 ± 0.090 ab |
| 75% LVC+ WS | 11.333 ± 1.258 e | 4.120 ± 0.209 cd | 22.083 ± 1.551 b | 1.034 ± 0.070 b | 1.597 ± 0.257 b | 2.877 ± 0.125 c | 1.679 ± 0.427 bc |
| 100% LVC | 14.200 ± 2.138 cd | 3.867 ± 0.252 d | 15.703 ± 0.754 c | 0.260 ± 0.036 g | 1.093 ± 0.110 c | 3.947 ± 0.311 b | 1.940 ± 0.052 ab |
| 100% LVC+ WS | 13.450 ± 0.606 cde | 2.573 ± 0.459 e | 26.363 ± 1.061 a | 1.343 ± 0.316 a | 0.557 ± 0.067 d | 2.633 ± 0.153 c | 1.383 ± 0.104 cd |
| *p*-value | LVC: 0.000 WS: 0.000 LVCxWS: 0.054 | LVC: 0.000 WS: 0.000 LVCxWS: 0.433 | LVC: 0.000 WS: 0.000 LVCxWS: 0.000 | LVC: 0.283 WS: 0.000 LVCxWS: 0.000 | LVC: 0.000 WS: 0.000 VCxWS: 0.485 | LVC: 0.070 WS: 0.000 LVCxWS: 0.411 | LVC:0.003 WS: 0.000 LVCxWS: 0.321 |

* WS: water stress; ** LVC: liquid leachate obtained from vermicompost. Different letters indicate significant difference according to a Duncan's multiple range test (*p* < 0.05).

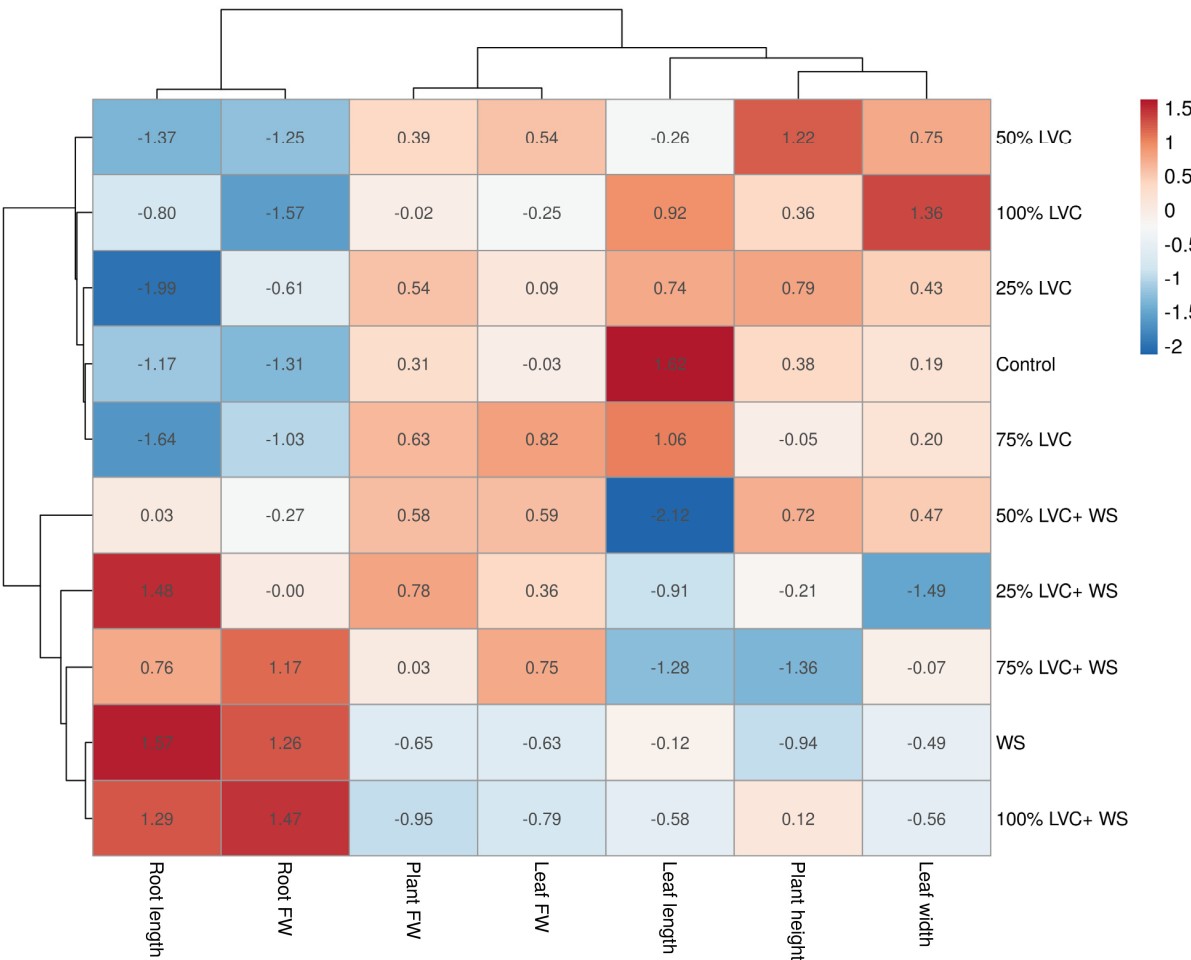

**Figure 1.** Heat map clustering of agronomic attributes corresponding to the treatments.

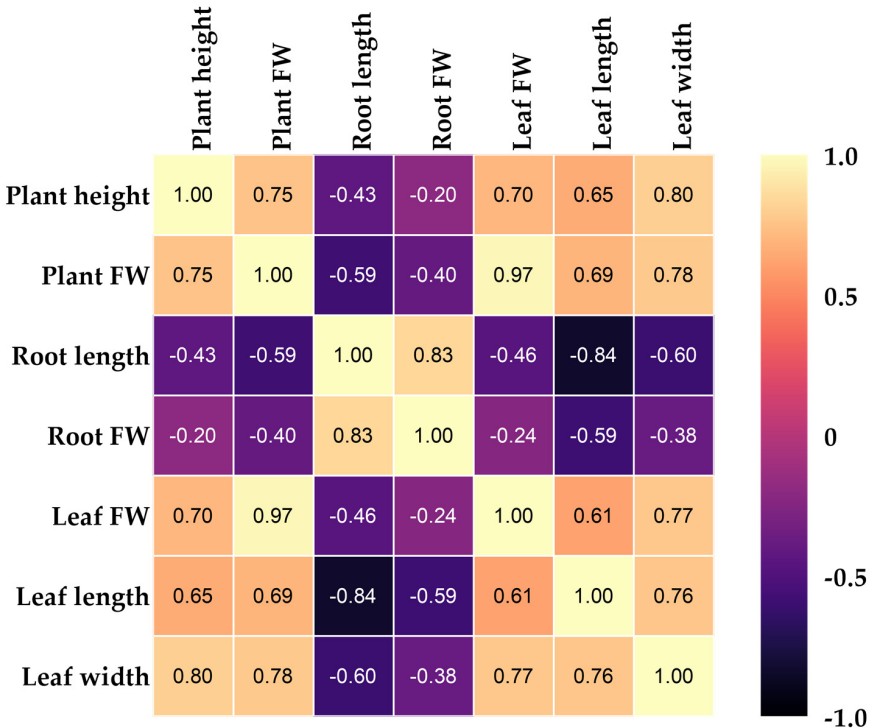

**Figure 2.** Correlation analysis of agronomic attributes corresponding to the treatments.

Furthermore, the agronomic attributes and treatments were discriminated on a biplot pair (Figure 3). Accordingly, two principal components with eigenvalues >1.0 accounted for 86.96% of the variability of the original data. Such a high explained variance suggests that principal component analysis can be a significant predictor in the assessment of relevant dependent variables corresponding to the treatments performed. The first principal component, $PC_1$ (*eigenvalue*: 4.788), accounting for 68.40% of the total variation, exhibited significant positive correlations with the plant height, plant FW, leaf FW, leaf length, and leaf width. On the other hand, in the second principal component, $PC_2$ (*eigenvalue*: 1.298), accounting for 18.56% of the total variation, root length and root FW had higher eigenvectors.

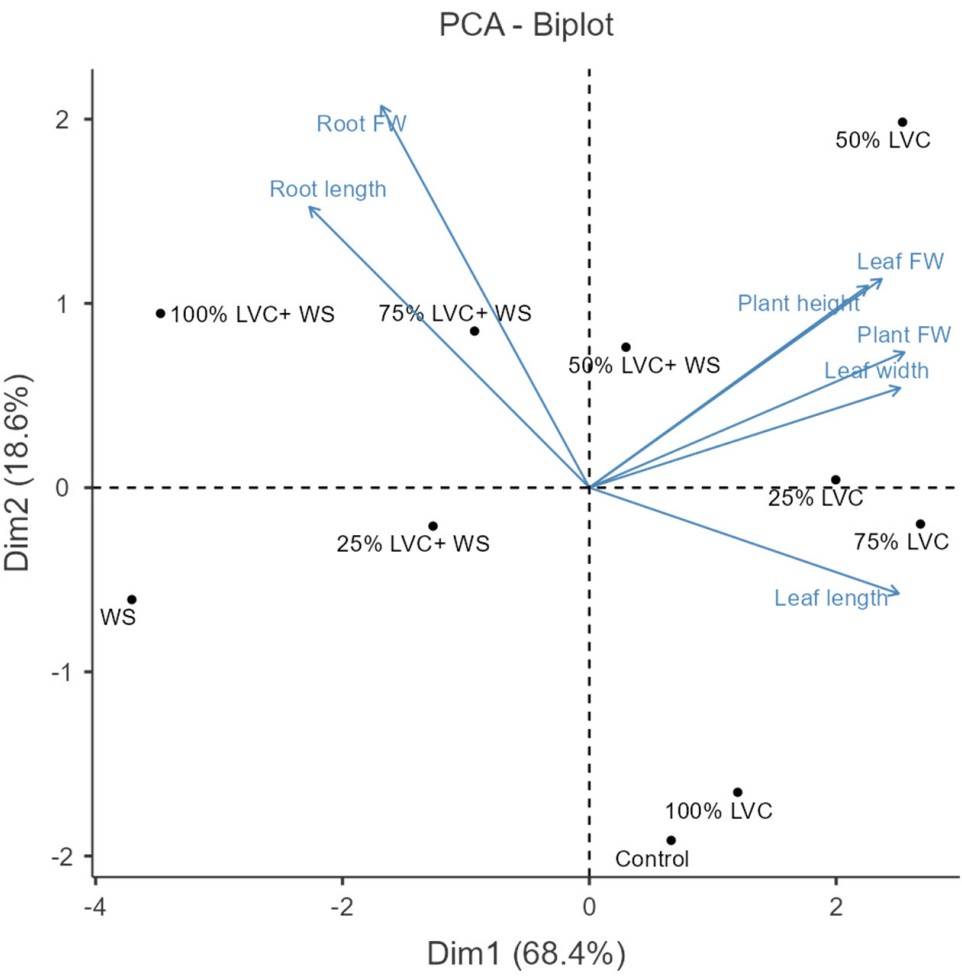

**Figure 3.** Principal component analysis of agronomic attributes corresponding to the treatments.

We finally performed a network plot analysis to reveal the link between individual vermicompost concentrations and water stress based on their performance on agronomic attributes (Figure 4). The plot consists of nodes via lines, and the depth of the line reveals the relation among the experimental groups. The thinner/lighter line presents the weaker relation whilst the thicker line shows the strong relations with each other. According to the network plot analysis, WS and WS + 100% LVC were clearly separated from the other treatments. WS + 100% LVC exhibited a similar performance to WS. Based on the network and thickness of the lines, it was clear that, based on the responses of the aforementioned agronomic attributes, the vermicompost applications without stress conditions were closely associated with each other, whereas the vermicompost-stress-interacting groups were also scattered close to each other.

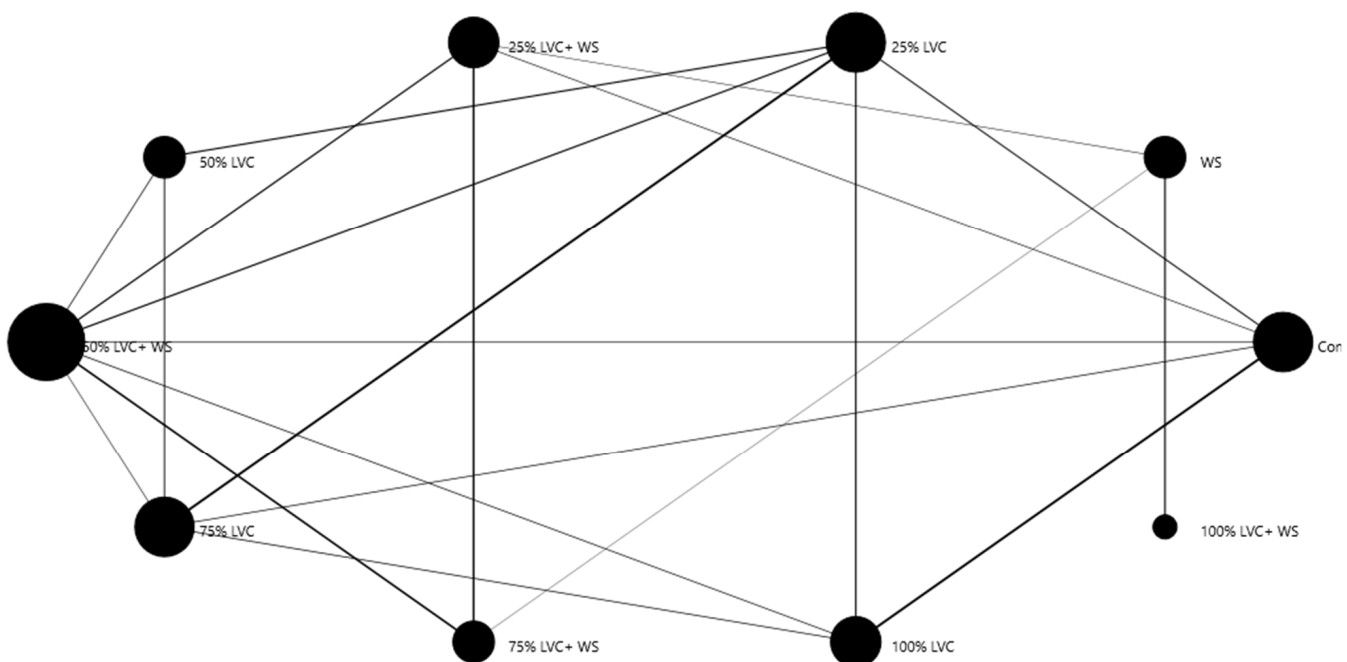

**Figure 4.** Network plot analysis of agronomic attributes corresponding to the treatments.

### 3.3. Essential Oil Compounds

The essential oil compounds identified in the basil leaves are presented in Table 3, following their elution orders on the HP-5 column. Of the identified compounds, estragole (methyl chavicol), eucalyptol, anethole, caryophyllene, humulene, germacrene D, and bisabolene were the predominant compounds. According to the statistical analysis, it can be observed that water stress, vermicompost, and their interactions significantly affect the percentage of the compounds ($p < 0.05$). In comparison to the control, water stress did not significantly affect the percentage of estragole. Regarding vermicompost treatments, neither 25% LVC nor 50% LVC critically affected the percentage value, but neither 75% LVC nor 100% LVC significantly decreased the percentage of estragole in well-watered basil plants. With respect to the vermicompost and water stress interactions, the highest percentage of estragole was noted at 25% LVC + WS, and the percentage decreased by higher concentrations of vermicompost and water stress interactions.

Water stress conditions reduced the percentage of eucalyptol, but 25% LVC increased the percentage value. However, higher concentrations of LVC significantly decreased the percentage of the compound in well-watered basil plants. In plants suffering from water stress conditions, vermicompost treatments, except 100% LVC, did not result in critical changes in the percentage of the compound present in comparison to the well-watered plants. Of the minor compound identified in the study, all increased with water stress conditions as well as interactions of 25% LVC + WS.

**Table 3.** Effects of liquid leachate obtained from vermicompost (25, 50, 75, and 100%) on essential oil compounds in basil leaves under water stress conditions (%).

| Treatments | Eucalyptol | Estragole | Anethole | Caryophyllene | Humulene | Germacrene D | Bisabolene |
|---|---|---|---|---|---|---|---|
| Control | 2.37 ± 0.08 b | 89.12 ± 2.10 b | 0.32 ± 0.06 d | 0.51 ± 0.08 e | 0.23 ± 0.04 e | 0.20 ± 0.04 g | 0.33 ± 0.05 e |
| WS * | 1.75 ± 0.18 ef | 87.77 ± 0.44 bc | 0.44 ± 0.06 c | 0.64 ± 0.06 de | 0.26 ± 0.03 e | 0.29 ± 0.01 f | 0.54 ± 0.06 de |
| 25% LVC ** | 2.04 ± 0.05c d | 88.33 ± 0.72 b | 0.31 ± 0.03 d | 0.69 ± 0.05 d | 0.53 ± 0.07 c | 0.69 ± 0.01 c | 0.94 ± 0.22 ab |
| 25% LVC +WS | 2.20 ± 0.12 bc | 91.28 ± 0.85 a | 0.65 ± 0.05 b | 0.91 ± 0.05 bc | 0.68 ± 0.01 b | 0.60 ± 0.02 d | 0.92 ± 0.02 abc |
| 50% LVC | 1.86 ± 0.04 de | 87.58 ± 0.65 bc | 0.41 ± 0.01 c | 0.86 ± 0.04 c | 0.40 ± 0.01 d | 0.52 ± 0.04 e | 0.96 ± 0.05 ab |
| 50% LVC +WS | 2.11 ± 0.15 c | 89.06 ± 0.16 b | 0.77 ± 0.04 a | 1.03 ± 0.05 ab | 0.83 ± 0.04 a | 0.59 ± 0.02 d | 0.76 ± 0.10 bcd |
| 75% LVC | 1.83 ± 0.06 def | 86.29 ± 0.59 cd | 0.46 ± 0.01 c | 0.91 ± 0.02 bc | 0.38 ± 0.02 cd | 0.82 ± 0.03 b | 1.00 ± 0.14 ab |
| 75% LVC +WS | 2.37 ± 0.05 b | 87.49 ± 0.46 bc | 0.64 ± 0.04 b | 0.94 ± 0.03 bc | 0.73 ± 0.04 b | 0.61 ± 0.02 d | 0.69 ± 0.14 cd |
| 100% LVC | 1.63 ± 0.08 f | 84.42 ± 0.74 d | 0.56 ± 0.04 b | 1.11 ± 0.15 a | 0.35 ± 0.03 d | 0.90 ± 0.01 a | 1.03 ± 0.05 a |
| 100% LVC +WS | 2.96 ± 0.05 a | 85.20 ± 0.32 d | 0.57 ± 0.04 b | 0.89 ± 0.05 bc | 0.58 ± 0.02 c | 0.62 ± 0.02 d | 0.82 ± 0.06 abc |
| *p*-value | LVC: <0.004 WS: <0.001 LVCxWS: <0.001 | LVC: <0.001 WS: <0.017 LVCxWS: 0.035 | LVC: <0.001 WS: <0.001 LVCxWS: <0.001 | LVC: <0.001 WS: <0.046 LVCxWS: <0.001 | LVC: <0.001 WS: <0.001 LVCxWS: <0.001 | LVC: <0.001 WS: <0.001 LVCxWS: <0.001 | LVC: <0.001 WS: <0.032 LVCxWS: <0.001 |

* WS: water stress; ** LVC: liquid leachate obtained from vermicompost. Different letters indicate significant difference according to a Duncan's multiple range test (*p* < 0.05).

*3.4. Heat Map Clustering, Correlation, Principal Component, and Network Plot Analyses of the Essential Oil Compound Corresponding to the Treatments*

Heat map clustering revealed two clusters of essential oil compounds corresponding to the treatments performed (Figure 5). Considering the treatments, as in the case of agronomic traits, groups that were well-watered and submitted to stress were clearly separated into distinct clusters. Of the identified compounds, caryophyllene, germacrene D, and bisabolene were grouped into the first cluster, while eucalyptol, estragole, anethole, and humulene were scattered into the second cluster. Estragole was dominant and its values peaked at a solo water stress (WS) level and 25% LVC + WS treatments. Interestingly, the predominant compounds, estragole and eucalyptol, were not correlated with any compounds ($p > 0.05$) according to the correlation analysis (Figure 6). In addition, essential oil compounds and treatments were scattered on a biplot pair via PCA (Figure 7). Accordingly, two principal components with eigenvalues > 1.0 accounted for 75.89% of the variability of the original data. The first principal component, $PC_1$ (*eigenvalue*: 3.33), accounting for 47.57% of the total variation, exhibited significant positive correlations with caryophyllene, germacrene D, and bisabolene. The second principal component, $PC_2$ (*eigenvalue*: 1.298), accounting for 28.32% of the total variation, was related to eucalyptol, estragole, anethole, and humulene. Regarding the network plot analysis (Figure 8), the linkage of individual vermicompost concentrations and water stress with each other based on their effect on essential compounds (eucalyptol, estragole, anethole, caryophyllene, humulene, germacrene D, and bisabolene) was established. According to the network plot analysis, it was clear that, based on the responses of the aforementioned essential oil compounds, the vermicompost applications without being subjected to stress were closely associated with each other, whereas the vermicompost-stress-interacted groups were also scattered close to each other. Furthermore, the control and stress groups presented a strong association in correspondence with vermicompost and its stress interactions.

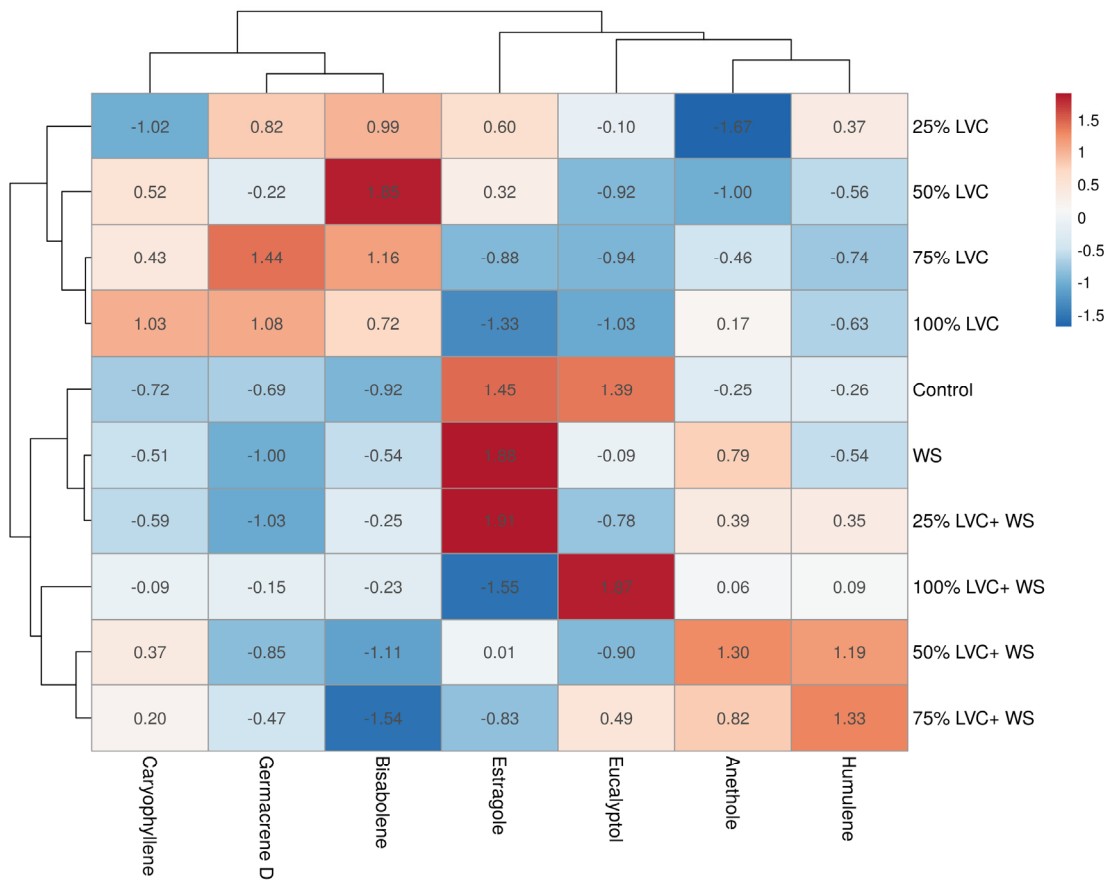

**Figure 5.** Heat map clustering of essential oil compounds corresponding to the treatments.

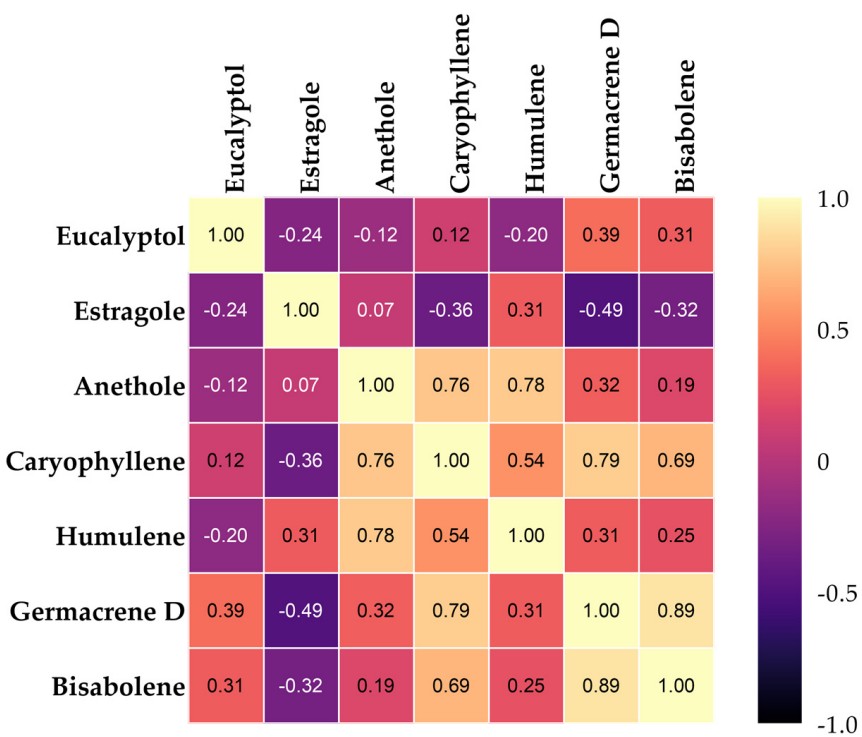

**Figure 6.** Correlation analysis of essential oil compounds corresponding to the treatments.

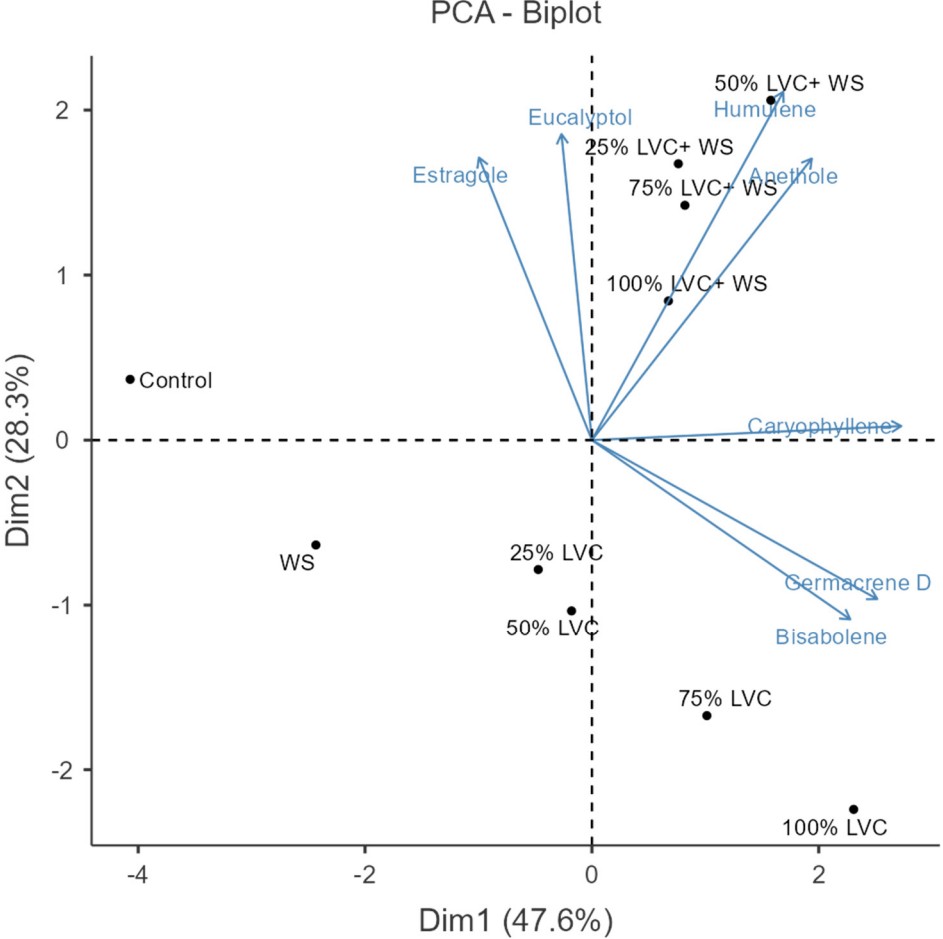

**Figure 7.** Principal component analysis of essential oil compounds corresponding to the treatments.

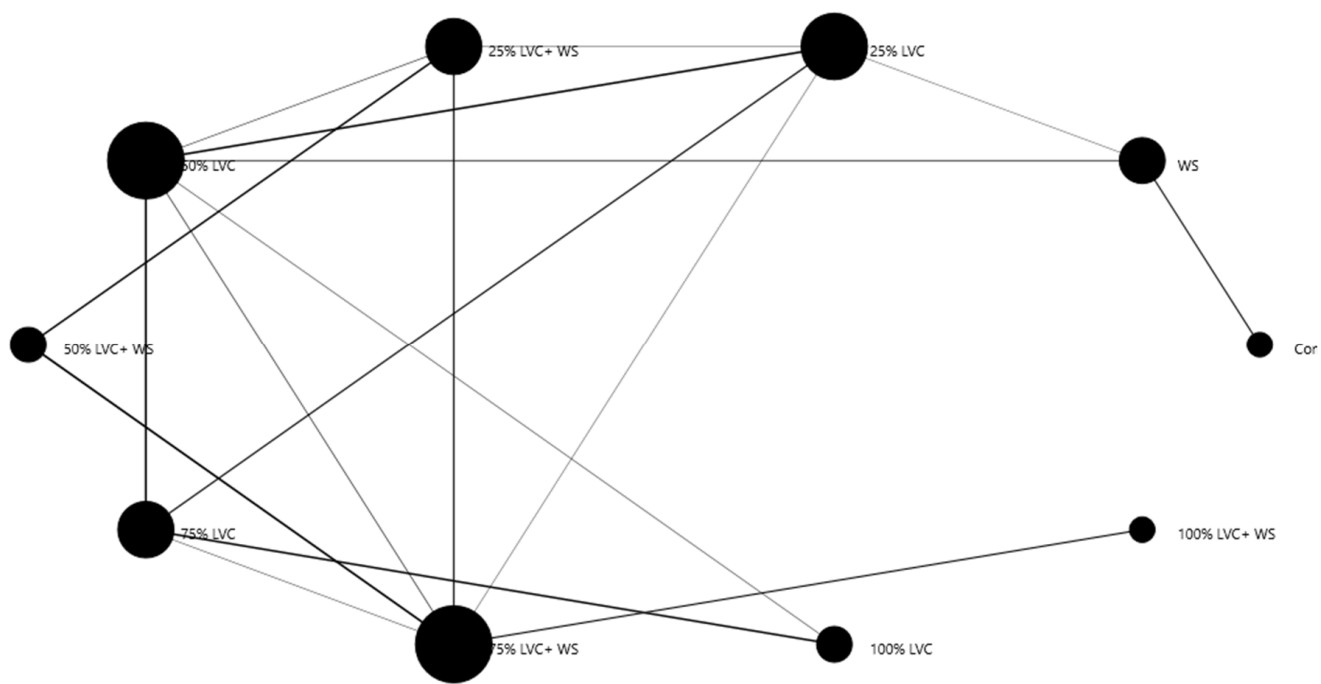

**Figure 8.** Network plot analysis of essential oil compounds corresponding to the treatments.

*3.5. Phenolic Acids and Flavonoids*

According to the LC–MS/MS analysis of the phenolics, twenty-six phenolic compounds were identified and quantified in the leaf samples (Table 4). Of the identified compounds, ascorbic acid was not significantly affected by either main-treatment water stress ($p$ = 0.654); vermicompost ($p$ = 0.373); or the interaction ($p$ = 0.071). Shikimic acid content was critically affected by water stress ($p$ = 0.005) and vermicompost ($p$ = 0.007), but the interactive effects of the treatments were not significant ($p$ = 0.105) (Table 5). The content of the compound was increased by approximately two-fold by water stress conditions. In comparison to the control, 50% LVC increased the content by 25.19%, but other vermicompost treatments did not exhibit substantial effects on the shikimic acid content in well-watered basil plants. As previously noted, although the interactive effects were not significant, the interaction decreased the shikimic acid content in comparison to the solo water stress treatment.

Of the major compounds of the basil plant, caffeic acid was significantly affected by water stress ($p$ = 0.037), vermicompost ($p$ = 0.000), and the interaction of the treatments ($p$ = 0.005). For instance, water stress critically reduced the content. While 25 and 50% of vermicompost treatments increased the content, critical decreases were noted with the increasing concentrations of vermicompost. Considering the interaction, it was determined that 25, 50, and 75% vermicompost concentrations and stress interactions increased the quantity of the related compound. Interestingly, 100% vermicompost concentrations have been noted to significantly inhibit caffeic acid biosynthesis, regardless of its solo uses or its interaction with stress. Similar to the case of caffeic acid, rosmarinic acid content was also significantly responsive to the treatments (water stress; $p$ = 0.000; vermicompost; $p$ = 0.000; interaction of water stress and vermicompost; $p$ = 0.000).

Quercimeritrin was significantly affected by the main treatments and their interactions ($p$ = 0.000). Water stress increased the content by approximately two-fold. However, solo treatments of vermicompost reduced the content in comparison to the control. As in the case of interaction, 75% LVC + WS treatments peaked the content of the compound. Of the identified compounds, water stress critically increased the content ($p$ = 0.021), whereas vermicompost treatments were not significant predictors for the content ($p$ = 0.071). However, the interaction of the treatments was significant ($p$ = 0.016).

**Table 4.** Effects of liquid leachate obtained from vermicompost (25, 50, 75, and 100%) on phenolic acids and flavonoids in basil leaves under water stress conditions (ng/uL).

| Compounds | Control | 25% LVC ** | 50% LVC | 75% LVC | 10% LVC | WS * | 25% LVC + WS | 50% LVC + WS | 75% LVC + WS | 100% LVC + WS |
|---|---|---|---|---|---|---|---|---|---|---|
| Ascorbic acid | 105.61 ± 4.26 | 104.58 ± 5.38 | 107.06 ± 3.06 | 104.79 ± 5.13 | 114.23 ± 9.18 | 115.05 ± 10.61 | 107.03 ± 2.68 | 112.13 ± 7.60 | 103.86 ± 0.98 | 103.13 ± 2.77 |
| Shikimic acid | 451.56 ± 76.13 | 461.57 ± 76.95 | 565.73 ± 103.00 | 353.15 ± 47.74 | 356.80 ± 40.00 | 816.95 ± 333.28 | 756.45 ± 150.81 | 561.68 ± 83.68 | 442.51 ± 97.31 | 391.64 ± 77.30 |
| Gallic acid | 0.00 ± 0.00 | 0.00 ± 0.00 | 60.30 ± 19.71 | 53.40 ± 11.34 | 8.62 ± 0.84 | 12.73 ± 1.42 | 0.99 ± 0.20 | 3.74 ± 2.30 | 7.34 ± 0.32 | 7.88 ± 2.06 |
| Protocatechuic acid | 0.00 ± 0.00 | 0.34 ± 0.59 | 0.00 ± 0.00 | 0.00 ± 0.00 | 0.00 ± 0.00 | 11.06 ± 0.92 | 0.00 ± 0.00 | 0.00 ± 0.00 | 0.00 ± 0.00 | 0.00 ± 0.00 |
| Chlorogenic acid | 1.91 ± 0.53 | 10.40 ± 5.18 | 6.62 ± 0.74 | 1.32 ± 0.14 | 1.02 ± 0.16 | 5.00 ± 0.72 | 1.32 ± 0.25 | 1.20 ± 0.39 | 1.13 ± 2.53 | 5.91 ± 0.18 |
| 4-Hydroxy-benzaldehyde | 0.00 ± 0.00 | 0.00 ± 0.00 | 0.96 ± 0.85 | 0.00 ± 0.00 | 0.00 ± 0.00 | 0.33 ± 0.30 | 0.65 ± 0.57 | 0.06 ± 0.11 | 0.00 ± 0.00 | 0.62 ± 0.54 |
| Caffeic acid | 29.65 ± 3.47 | 87.91 ± 6.59 | 67.15 ± 20.07 | 14.39 ± 0.00 | 0.00 ± 0.82 | 8.12 ± 2.30 | 62.79 ± 12.62 | 73.92 ± 7.85 | 20.53 ± 0.00 | 0.00 ± 1.56 |
| Syringic acid | 125.91 ± 8.97 | 129.82 ± 9.16 | 127.10 ± 3.15 | 143.32 ± 1.21 | 154.89 ± 7.16 | 125.34 ± 7.88 | 117.19 ± 2.03 | 128.50 ± 1.72 | 136.23 ± 6.73 | 127.63 ± 2.70 |
| P-coumaric acid | 0.82 ± 0.76 | 0.00 ± 0.01 | 0.00 ± 0.00 | 0.00 ± 0.00 | 0.00 ± 0.00 | 0.86 ± 0.76 | 0.00 ± 0.00 | 0.00 ± 0.00 | 0.00 ± 0.0 | 0.00 ± 0.00 |
| Polydatine | 0.92 ± 0.27 | 0.07 ± 0.12 | 0.03 ± 0.06 | 0.00 ± 0.00 | 0.00 ± 0.00 | 0.16 ± 0.27 | 0.26 ± 0.31 | 0.00 ± 0.00 | 0.00 ± 0.00 | 1.16 ± 2.01 |
| Trans-ferulic acid | 103.48 ± 54.08 | 1722.96 ± 176.75 | 1614.05 ± 242.85 | 93.27 ± 8.62 | 68.85 ± 20.43 | 716.28 ± 70.83 | 2447.94 ± 475.02 | 151.43 ± 8.06 | 506.89 ± 98.17 | 34.65 ± 8.59 |
| Quercimeritrin | 87.16 ± 39.27 | 9.90 ± 1.41 | 48.59 ± 4.05 | 58.18 ± 7.49 | 41.38 ± 2.51 | 238.27 ± 14.54 | 245.87 ± 69.37 | 27.00 ± 5.29 | 497.84 ± 153.88 | 102.78 ± 8.98 |
| Cynarin | 24.54 ± 3.30 | 23.70 ± 1.49 | 24.87 ± 1.19 | 23.38 ± 1.68 | 22.82 ± 0.80 | 22.48 ± 1.52 | 23.20 ± 0.73 | 23.48 ± 1.88 | 23.89 ± 1.29 | 25.89 ± 1.83 |
| Hyperocide | 24.17 ± 11.03 | 5.16 ± 1.52 | 10.05 ± 7.35 | 27.74 ± 1.22 | 28.16 ± 7.99 | 295.74 ± 15.51 | 113.92 ± 4.30 | 32.17 ± 3.01 | 205.73 ± 5.84 | 34.52 ± 10.24 |
| Quercetin-3-glucoside | 10.73 ± 5.09 | 5.22 ± 2.88 | 8.66 ± 2.50 | 2.89 ± 0.41 | 9.31 ± 2.73 | 80.59 ± 12.89 | 39.86 ± 3.85 | 10.85 ± 1.26 | 53.93 ± 18.63 | 10.55 ± 3.49 |
| Rutin | 229.90 ± 12.22 | 231.50 ± 11.82 | 258.32 ± 13.01 | 218.27 ± 7.17 | 243.24 ± 10.60 | 354.55 ± 102.51y | 311.79 ± 70.89 | 223.30 ± 1.48 | 240.65 ± 5.30 | 235.38 ± 4.72 |
| Isoquercitrin | 11.01 ± 5.21 | 6.38 ± 1.69 | 8.05 ± 2.47 | 2.65 ± 0.45 | 10.29 ± 0.68 | 73.27 ± 17.35 | 38.05 ± 2.37 | 10.61 ± 0.71 | 56.19 ± 19.92 | 10.25 ± 3.41 |
| Resveratrol | 5.32 ± 1.14 | 7.32 ± 2.36 | 4.33 ± 2.34 | 9.15 ± 2.35 | 8.22 ± 2.14 | 5.32 ± 1.00 | 10.94 ± 1.26 | 7.39 ± 0.99 | 7.93 ± 0.70 | 10.42 ± 1.19 |
| Naringin | 313.50 ± 18.35 | 296.67 ± 12.64 | 283.66 ± 5.77 | 313.50 ± 5.41 | 299.47 ± 14.59 | 294.55 ± 5.98 | 325.34 ± 9.34 | 297.84 ± 11.80 | 304.97 ± 15.94 | 305.51 ± 11.09 |
| Rosmarinic acid | 682.88 ± 285.372 | 5950.13 ± 110.82 | 11251.46 ± 1230.71 | 352.23 ± 75.79 | 4690.59 ± 276.48 | 13904.23 ± 2.460.64 | 29879.68 ± 600.89 | 11319.82 ± 1149.15 | 1798.03 ± 446.64 | 5192.40 ± 329.97 |
| Neohesperidin | 8.68 ± 1.51 | 2.98 ± 0.83 | 6.20 ± 0.95 | 0.00 ± 0.00 | 17.39 ± 4.04 | 13.34 ± 1.02 | 31.12 ± 1.83 | 23.06 ± 2.58 | 48.92 ± 10.52 | 5.36 ± 0.77 |
| Ellagic acid | 34.67 ± 8.81 | 0.00 ± 0.00 | 0.00 ± 0.00 | 0.00 ± 0. 00 | 0.00 ± 0.00 | 12.82 ± 2.33 | 0.00 ± 0.00 | 0.00 ± 0.00 | 0.00 ± 0.00 | 35.63 ± 8.11 |
| Naringenin | 64.78 ± 6.840484 | 72.23 ± 11.57 | 38.51 ± 17.38 | 20.28 ± 1.71 | 18.05 ± 2.98 | 6.41 ± 0.93 | 18.19 ± 1.21 | 17.65 ± 6.71 | 14.67 ± 1.27 | 22.38 ± 9.79 |
| Silibinin | 10.53 ± 0.60 | 10.59 ± 0.58 | 10.62 ± 0.55 | 11.06 ± 1.06 | 10.63 ± 0.32 | 10.50 ± 0.66 | 10.50 ± 0.62 | 10.74 ± 0.46 | 10.63 ± 0.37 | 10.80 ± 0.38 |
| 3-Hydroxyflavone | 21.20 ± 1.83 | 20.50 ± 1.72 | 22.48 ± 2.15 | 19.69 ± 1.29 | 21.57 ± 3.00 | 23.55 ± 2.12 | 22.31 ± 3.56 | 20.79 ± 1.22 | 19.73 ± 3.27 | 21.63 ± 2.07 |
| Diosgenin | 5.15 ± 2.91 | 2.29 ± 1.05 | 3.49 ± 0.39 | 4.21 ± 0.86 | 4.10 ± 0.62 | 4.17 ± 0.31 | 2.23 ± 0.72 | 2.08 ± 0.27 | 1.60 ± 0.72 | 6.40 ± 1.80 |

* WS: water stress; ** LVC: liquid leachate obtained from vermicompost. Different letters indicate significant difference according to a Duncan's multiple range test ($p < 0.05$).

**Table 5.** Variance analysis of phenolic compounds corresponding to the treatments.

| Compounds | Water Stress | Vermicompost | Water Stress × Vermicompost |
|---|---|---|---|
| Ascorbic acid | 0.654 [ns] | 0.373 [ns] | 0.071 [ns] |
| Shikimic acid | 0.005 | 0.007 | 0.105 [ns] |
| Gallic acid | 0.000 | 0.000 | 0.000 |
| Protocatechuic acid | 0.000 | 0.000 | 0.000 |
| Chlorogenic acid | 0.063 [ns] | 0.006 | 0.000 |
| 4-Hydroxybenzaldehyde | 0.326 [ns] | 0.226 [ns] | 0.012 |
| Caffeic acid | 0.037 | 0.000 | 0.005 |
| Syringic acid | 0.000 | 0.000 | 0.000 |
| P-coumaric acid | 0.946 [ns] | 0.001 | 1.000 [ns] |
| Polydatine | 0.645 [ns] | 0.383 [ns] | 0.199 [ns] |
| Trans-ferulic acid | 0.454 ns | 0.000 | 0.000 |
| Quercimeritrin | 0.000 | 0.000 | 0.000 |
| Cynarin | 0.906 [ns] | 0.844 [ns] | 0.126 [ns] |
| Hyperocide | 0.000 | 0.000 | 0.000 |
| Quercetin-3-glucoside | 0.000 | 0.000 | 0.000 |
| Rutin | 0.021 | 0.071 [ns] | 0.016 |
| Isoquercitrin | 0.000 | 0.000 | 0.000 |
| Resveratrol | 0.021 | 0.001 | 0.095 [ns] |
| Naringin | 0.335 [ns] | 0.061 ns | 0.020 |
| Rosmarinic acid | 0.000 | 0.000 | 0.000 |
| Neohesperidin | 0.000 | 0.000 | 0.000 |
| Ellagic acid | 0.065 [ns] | 0.000 | 0.000 |
| Naringenin | 0.000 | 0.000 | 0.000 |
| Silibinin | 0.820 [ns] | 0.872 [ns] | 0.918 [ns] |
| 3-Hydroxyflavone | 0.558 [ns] | 0.404 [ns] | 0.599 [ns] |
| Diosgenin | 0.229 [ns] | 0.001 | 0.028 |

ns: non-significant.

*3.6. Heat Map Clustering, Correlation, Principal Component and Network Plot Analyses of the Phenolics and Flavonoids Corresponding to the Treatments*

According to the heat map clustering (Figure 9), it can be observed that two distinct clusters were obtained in relation to the treatments. The first cluster was composed of 75% LVC, control, and 75% LVC + WS, while the other treatments were grouped under the second cluster. However, the stress or non-stress groups were not well-discriminated. With respect to the phenolic compounds, the first cluster included ascorbic acid, syringic acid, shikimic acid, rutin, and naringin. Those compounds attained their highest content values in treatments of 75% LVC. The major compound of basil plants, rosmarinic acid, was grouped in the second cluster. Contrary to the compounds in the first cluster, the lowest content of rosmarinic acid was recorded in treatments of 75% LVC. According to the correlation analysis (Figure 10) of major compounds (caffeic and rosmarinic acids), caffeic acid was only significantly correlated with trans-ferulic acid ($r = 0.667$; $p < 0.05$) and diosgenin ($r = -0.638$; $p < 0.05$). Similar to the case of the major essential oil compound (estragole), the major phenolic compound (rosmarinic acid) was not significantly correlated with any phenolic compounds ($p > 0.05$). In addition, we performed PCA analysis to scatter the phenolic compounds and treatments on a biplot pair (Figure 11). Accordingly, two principal components ($PC_1$ *eigenvalue*: 8.93 (89.31%) and $PC_2$ *eigenvalue*: 0.92 (9.20%)) accounted for 98.51% of the variability of the original data. According to the loading factors, shikimic acid, naringin, and rosmarinic acid were clearly separated from the other compounds on the biplot pair. Similar to the cases concerned with the relations of the experimental groups and their performance phenolic compounds, the network plot analysis revealed the only relations among 25% LVC, 75% LVC, 100% LVC, and 50% LVC + WS (Figure 12).

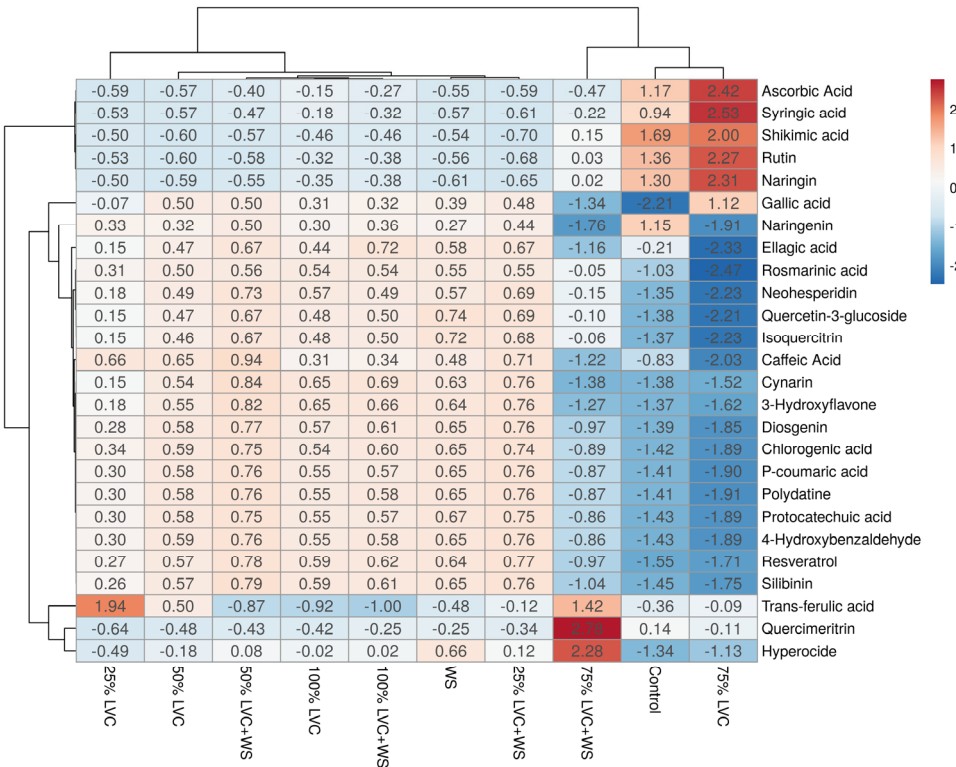

**Figure 9.** Heat map clustering of phenolics and flavonoids corresponding to the treatments.

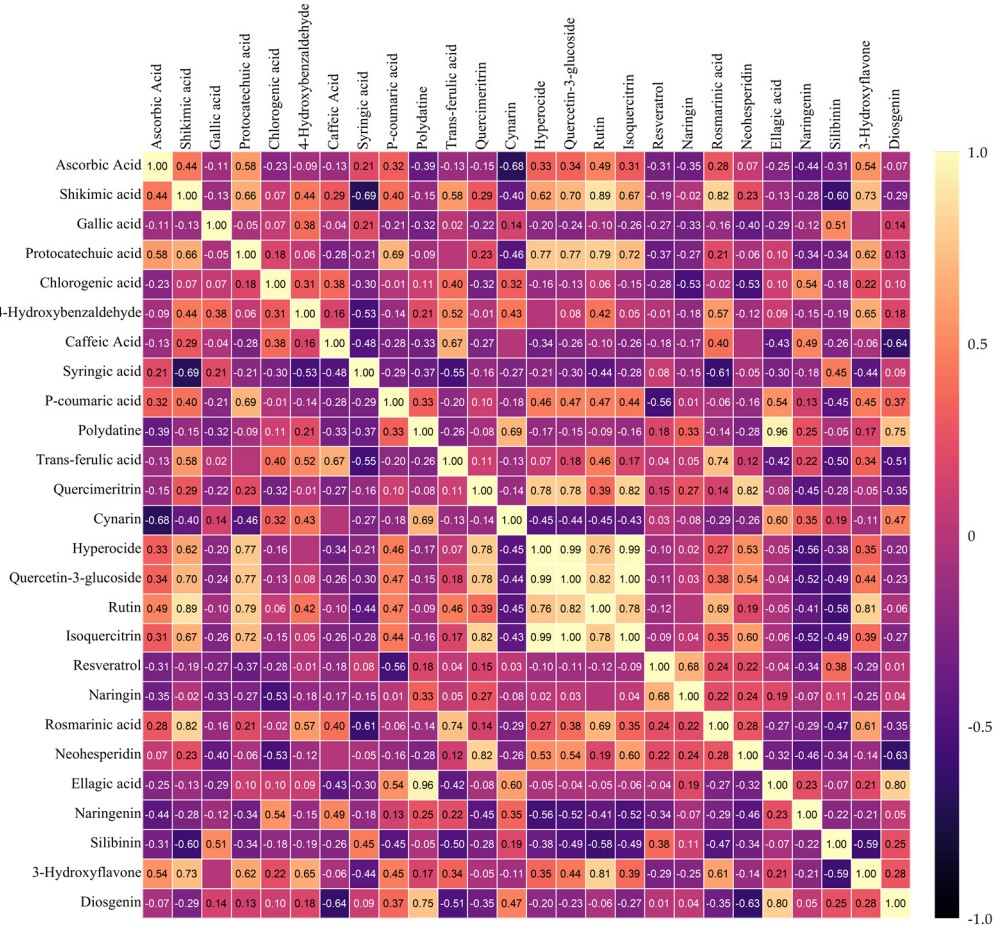

**Figure 10.** Correlation analysis of phenolics and flavonoids corresponding to the treatments.

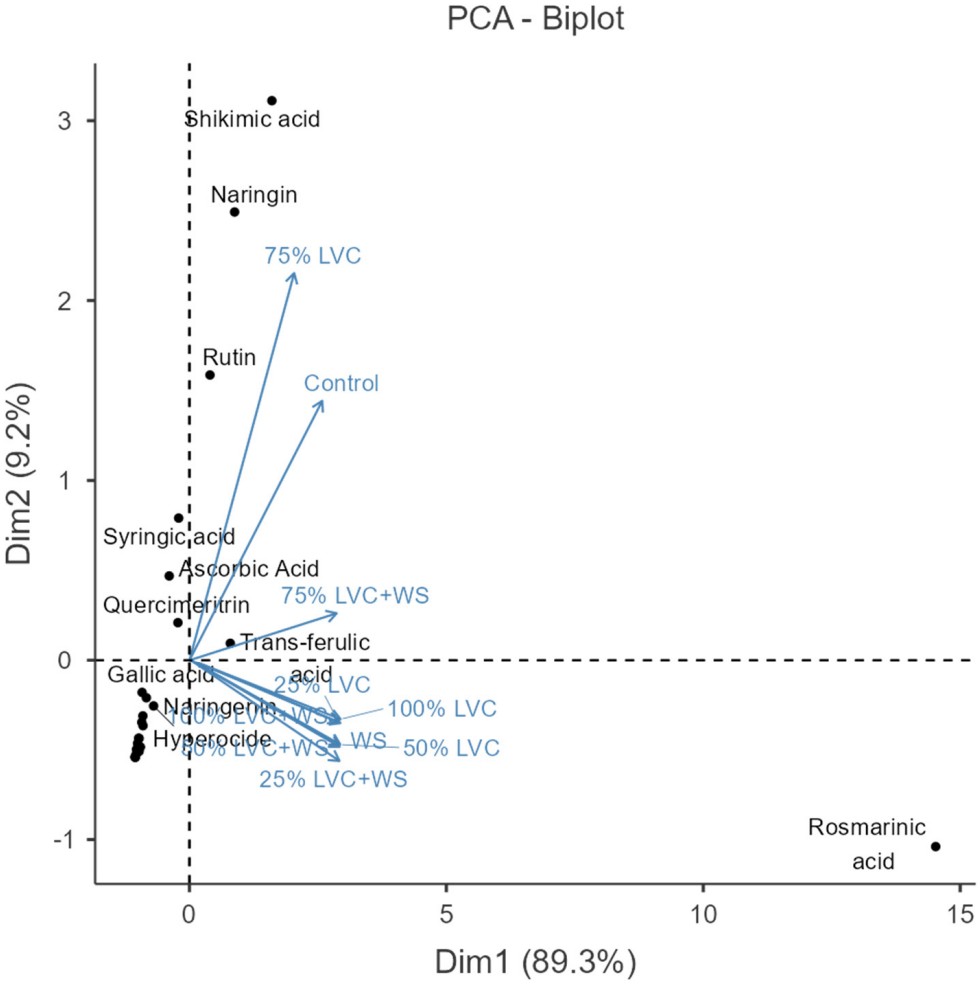

**Figure 11.** Principal component analysis of phenolics and flavonoids corresponding to the treatments.

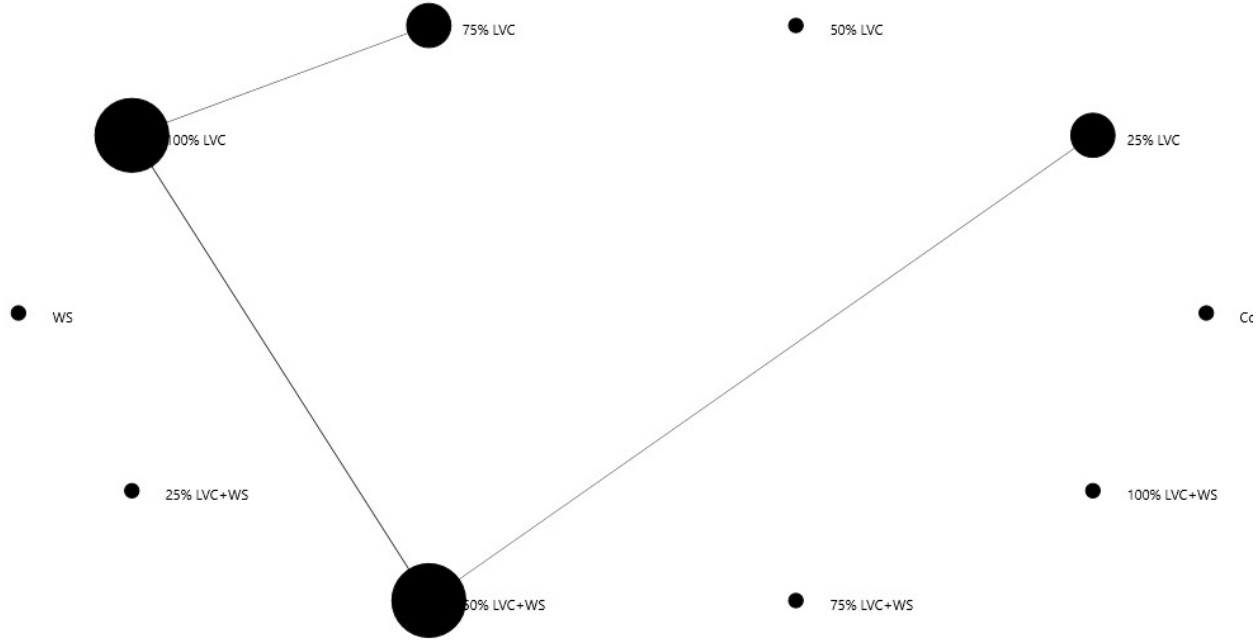

**Figure 12.** Network plot analysis of phenolics and flavonoids corresponding to the treatments.

## 4. Discussion

Water stress is one of the most important environmental problems that causes drastic problems in both developed and developing countries. After conducting a basic search on SCOPUS with criteria, including "drought stress OR water stress" on 20 July 2022, 253.552 documents obtained from SCOPUS were recorded. Considering the devastating effects of water stress, the direct effects of the stress were manifested as retarded plant growth and performance as well as a loss of crop productivity, in particular [40]. For that reason, intensive research has been conducted in order to cope with drought stress and to understand the action mechanism of droughts on crop and non-crop plant species. Just as the effects of limited water supply are not the same for all plant species, the duration, severity, frequency, and period of occurrence of water stress cause critical effects on each plant species [4,41]. These results, in a way, limit our ability to make general postulations about the mechanisms of drought. For these reasons, this deserves further investigations in the relevant fields. In an attempt to combat water stress conditions, due to fact that breeding studies are a process requiring many years and a lot of expertise, short-term solutions, such as chemical inputs in agricultural fields, have been suggested. However, the long-term and excessive chemical fertilizer input causes serious negative effects on plants and the ecosystem [42]. For these reasons, the demand for organic-sourced fertilizers, which produce fewer negative effects on the natural environment, has recently increased [43–45]. Although organic fertilizers can be obtained from quite varying sources, vermicompost amendments are the most commonly and recently used organic fertilizers [30,46–49]. It should be emphasized that the related studies mostly focus on the development and productivity of plants. Although this varies according to the concentration and application time, the positive effect of vermicompost on the growth and development of plants has been clearly demonstrated in the research. In our previous study [30], we tested the potential effects of a solid form of vermicompost on basil plants subjected to water stress conditions. Being very similar to the present study, we monitored the changes in the agronomic traits and secondary metabolites of basil under water stress conditions. As an effective approach to waste management, in this study, the liquid leachate obtained from vermicompost was assayed for its potential effects in basil against water stress conditions. Based on the analysis regarding the physico-chemical composition of liquid leachate, contrary to the solid form of vermicompost, leakage was observed to be poor in relation to the organic content and other elements. However, the microbial composition of the vermicompost was not analyzed in our study. As the effects of vermicompost are not only dependent on the organic and element content, but also on hormone-like compounds and the microbial composition [50–52], we hypothesized that liquid leachate might also be a critical discriminative factor and predictor in buffering the adverse impacts of basil plants submitted to water stress conditions.

Water stress caused critical damage to the agronomic traits of basil. These results are consistent with the previous reports on basil plants suffering from water stress conditions [7,8,30]. However, the root systems of the plants might be positively affected by the decline in water levels in the soil [30]. The significant increases in both the root length and root FW were noted and those parameters were positively correlated ($r = 0.83$; $p < 0.05$). On the other hand, as expected, positive results were obtained for the agronomic properties of the basil plant with liquid vermicompost applications independent of stress conditions. Similar to the stress-suffered basil plants, the vermicompost amendment affirmatively influenced the under-ground components of the basil plants. Pant et al. [53] also reported that plant and root growth as well as overall crop productivity were achieved with the vermicompost. These results might suggest that the root system of a plant could be an important distinguishing and predictive factor. However, basil leaves are significant in view of industrial demand. For that reason, vermicompost treatments might not be efficient to combat stress in basil plants, but these applications may be more realistic and effective for plants evaluated for their underground parts. In general, the augmented vegetative growth attributes of plants exposed to vermicompost were explained by the enrichment of

the growth media in terms of both nutrients and organic matter [30,53,54]. Additionally, previous reports revealed that the foliar application of vermicompost leakage enhanced the photosynthesis efficiency in either control plants or those submitted to stress conditions [55]. Although both organic and nutritional element contents were low according to the vermicompost fertilizer analysis, the trial soil might be enriched with the enzymes or hormone-like substances of the relevant fertilizer. Depending on the enrichment, additional microorganism, enzyme, and hormone inputs might be added to the soil structure, which in turn might directly contribute to the vegetative growth of the plants [51,52]. As clearly reported by [56,57], the compounds with molecular structure analogues similar to auxin and cytokinin were available in the compost. In this study, we did not measure the phytohormones or plant growth-stimulating compounds. However, we can suggest their potential and plausible roles in plant responses. Furthermore, the plant growth might be linked to the absorption of elements through the plasmatic membrane H+-ATPase-aided activation of macro- or micro-nutrient uptakes [50].

Basil plants are reputed medicinal and aromatic plants due to their secondary metabolites (terpenoids and phenolic compounds). The alterations in the patterns of their metabolites as a response to water stress conditions have been reported in numerous studies [8,30,58–60]. However, the interactions of liquid vermicompost and water stress have not been investigated hitherto. It has been widely reported that the excess carbon that is not used in growth and development processes (primary metabolism) in plant systems is used in secondary metabolism. The shift of carbon surplus from primary to secondary metabolism is one of the critical defense strategies used by plants against stress conditions. In addition to the enzymatic antioxidant system, plants have also developed a non-enzymatic defense system with the construction of secondary metabolites [41,61–64]. According to the current results, however, the major compound, estragole percentage, was not critically affected by water stress, in comparison to the control, but the percentage peaked at the interaction of 25% LVC and water stress. On the other hand, eucalyptol percentage was significantly decreased in water stress-submitted plants, in comparison to the control, but the percentage of the compound reached the highest value at the interaction of 100% LVC and water stress. The decline in the content of both compounds was also previously reported in basil plants exposed to water stress conditions [30].

Phenolic compounds are one of the remarkable groups of metabolites acknowledged for their critical roles in reducing the oxidative stress, being reported in numerous studies [65–67]. However, the studies conducted with respect to the plasticity of phenolic compounds as a response to vermicompost and interaction of vermicompost with water stress are quite limited. Of those reports, Celikcan et al. [30] assayed the solid form of vermicompost for basil plants against water stress conditions. The major phenolic compounds of basil plant are caffeic, rosmarinic, and chicoric acids [68]. In the present study, the contents of caffeic and rosmarinic acids were quantified and significant decreases were noted in rosmarinic acid content in relation to water stress conditions, in accordance with the results obtained from previous studies [69–71]. Additionally, water stress critically reduced the caffeic acid content. In addition, it has been noted that regardless of stress conditions, 100% vermicompost concentrations used alone or independent of their interaction with stress significantly inhibited caffeic acid biosynthesis. However, it was observed that lower concentrations of liquid vermicompost applications increased the quantity of the related compound interacting with stress conditions. However, solid vermicompost forms, drought stress, and their interactions significantly increased the caffeic acid content of the basil plant [30]. These differences might be explained by the physico-chemical composition of vermicompost and the type as well as duration of stress factors, since other factors, *viz.*, basil cultivars, growing media, or stress-timing, were the same. Previous reports have revealed that organic amendment positively affected the quantity of total phenolic content [72]. In this study, we profiled the quantities of phenolic acids and flavonoids instead of total phenolic content of the basil plants as a response of vermicompost and

its interaction with stress. However, the enrichment of the growing media with organic fertilizer critically affected the phenolic acids [73].

## 5. Conclusions

Water stress, as expected, critically resulted in reductions in agronomic attributes, such as plant height, plant fresh weight, root fresh weight, leaf length, and leaf diameter. Despite the water stress conditions, enriching the growth media with liquid leakage obtained from vermicompost crucially affected the agronomic attributes of well-watered basil plants. In particular, the highest values with respect to the above-ground parts were observed at a 50% concentration, whilst the highest values of under-ground parts were recorded at a 100% concentration of leakage. Considering the interactions of water stress and vermicompost, however, the interaction only had significant effects on the root length and root fresh weight. Regarding the major essential oil compound (estragole), the highest estragole content was determined in the 25% vermicompost + water stress, water stress, and control groups. Of the major phenolic compounds, caffeic acid decreased as a result of water stress but increased with the vermicompost treatments. The rosmarinic acid content increased as a result of water stress, reaching the highest content at 25% vermicompost and water stress interaction. In general, 25% and 50% vermicompost applications increased the content of phenolic compounds in plants under either well-watered or stress conditions. To the best of our knowledge, the present study is one of the first studies of its kind to analyze essential oil, phenolic acid, and flavonoids present in basil plants submitted to water stress conditions.

**Author Contributions:** Conceptualization, M.K.; methodology, M.K. and M.Z.K.; software, M.K. and M.G.K.; validation, M.K.; formal analysis, M.K.; investigation, H.K., M.Z.K., F.C. and M.G.K.; data curation, H.K., M.Z.K. and F.C.; writing—original draft preparation, M.K.; writing—review and editing, M.K.; visualization, M.K. and M.G.K.; supervision, M.K.; project administration, M.K. and M.Z.K.; funding acquisition, M.K. All authors have read and agreed to the published version of the manuscript.

**Funding:** The study was financially supported by the project coordination unit of Igdir University (Türkiye) with project number: TBY1220Y18. In this regard, we would like to send our deep thanks to Igdir University.

**Data Availability Statement:** The data used to support the findings were all included in this study.

**Acknowledgments:** The present study was derived from the Master's thesis of H. Kosem (supervised by M. Kulak).

**Conflicts of Interest:** The authors declare that they have no known competing financial interest or personal relationship that could have appeared to influence the work reported in this paper.

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
