# Peer review of "Liquid Leachate Produced from Vermicompost Effects on Some Agronomic Attributes and Secondary Metabolites of Sweet Basil (Ocimum basilicum L.) Exposed to Severe Water Stress Conditions"

_horticulturae, doi:10.3390/horticulturae8121190_

Round 1
Reviewer 1 Report
In the manuscript titled „Liquid leachate from vermicopost effects on some agronomic attributes and secondary metabolites of sweet basil (Ocimum basilicum L.) exposed to severe water stress“ authors dealt with a significant topic with potential practical applications. However, the manuscript requires significant corrections and refinements in order to be suitable for publication.
Although the authors investigated the effects of vermicompost on certain morphological and physiological characteristics of basil, taking into account the conditions of water deficit and stress, it is necessary to add additional measurements (such as MDA level, EC, concentrations of reactive species and enzymatic activity or photosynthetic parameters) in order to be able to consider severe water stress and to justify the conclusions about application of vermicompost under conditions of water stress.
Specific comments:
Hypothesis must be improved. Last part of the paragraph belongs to the Materials and Methods section.
Materials and Methods – Please provide additional information about sweet basil used in the study. “Green leaves” is common term, but it is insufficient for horticultural demands. There are numerous basil varieties and cultivars with green leaves.
It is unclear how authors watered basil plants. What was the water regime from germination until the stress and vermicopost application? Please explain what well-watered means.
L294 – “3.5. Individual phenolic acids” – according to the presented results authors analysed different phenolic compounds, i.e., phenolic acids and flavonoids. Please correct this here and through the whole text.
Table 4 and table 5 should be places in opposite order.
Figure 3a and b – please improve it, it is hard to read, or consider to move them into the supplementary material.
General comments:
Manuscript could be better written since in its current form there are a lot of unclear sentences like the following:
L19-20 „Considering the interactions of water stress and vermicopost; however…“
L39-40 „…due to ever increasing water supply problems“
L42 – „hampered growth and performance“
L55 – „…powerful attempts but involving chemical“
L85 – „augmented tolerance“
L366-367 – „about the effect mechanism of drought“
L379-381 – „In our previous report [30], we have assayed solid form of vermicompost in water-stress submitted basil plants and secondary metabolites of basil were, in addition to the agronomic attributes, revealed.“ - please reformulate
L388-389 „critical predictor for basil against water stress“ - it is not justified, please reformulate
L390-391 – „Water stress caused critical damage to the agronomic attributes of basil, being consistent with previous reports for basil, in particular” - please reformulate
L399-404 – What do you mean by these statements?
L426 – „elaborated defence strategies“ ???
Author Response
Dear Reviewer-1, please see the attachment.
Best regards

Reviewer 2 Report
The work is well planned and carried out using both classical methods of analysis and morpho-physiological parameters of basil plants, as well as using modern methods of quantification of phenolic compounds by LC-MS/MS.
The used literature corresponds to the set tasks and contains works of recent years.
The work corresponds to the direction of the journal and can be accepted for publication.
Author Response
Dear Reviewer-2, please see the attachment.
Best regards

Reviewer 3 Report
Dear authors,
Your manuscript requires some corrections that it could be public in Horticulture journal:
- in the paper there is nothing about the basil, why this plant species was chosen, what is its importance as a cultivated plant, what are its systematic characteristics, etc. Also, it is not emphasized which type of basil with green leaves was used, since there are several of them.
- in the last paragraph of the introductory part, the segment between lines 90 and 96 belongs more to the Material and methods part, so it needs to be moved
- it is unclear why the experiment was stopped on the 11th day, whether it is in accordance with earlier literature data
- Figures 1 and 2 are blurry, the text on them is hard to read
- the Tables are too voluminous, maybe they should be divided into several smaller tables
- it is necessary to correct the English, grammatically many sentences are incorrect, and unclear (for example, sentences on lines 39-40, 390-391, 399-404, 425-426 etc.)
Author Response
Dear Reviewer-3, please see the attachment.
Best regards

Round 2
Reviewer 1 Report
Authors improved the presentation of their results.